# Effects of Daidzein, Tempeh, and a Probiotic Digested in an Artificial Gastrointestinal Tract on Calcium Deposition in Human Osteoblast-like Saos-2 Cells

**DOI:** 10.3390/ijms25021008

**Published:** 2024-01-13

**Authors:** Iskandar Azmy Harahap, Anna Olejnik, Katarzyna Kowalska, Joanna Suliburska

**Affiliations:** 1Department of Human Nutrition and Dietetics, Faculty of Food Science and Nutrition, Poznan University of Life Sciences, 60-624 Poznan, Poland; iskandar.harahap@up.poznan.pl; 2Department of Biotechnology and Food Microbiology, Faculty of Food Science and Nutrition, Poznan University of Life Sciences, 60-624 Poznan, Poland; anna.olejnik@up.poznan.pl (A.O.); katarzyna.kowalska@up.poznan.pl (K.K.)

**Keywords:** isoflavones, probiotics, tempeh, calcium, Caco-2, Saos-2

## Abstract

Adequate calcium intake is crucial for the prevention and treatment of bone-related issues. Developing a nutritional source of readily bioavailable calcium is particularly significant for individuals deficient in this essential element and at risk of developing osteoporosis. This research aimed to evaluate the impact of tempeh (T), daidzein (D), and *Lactobacillus acidophilus* (LA) within a simulated intestinal environment consisting of Caco-2 epithelial and Saos-2 cells, focusing on their implications for bone mineralization mechanisms. In the initial phase, calcium bioaccessibility from calcium citrate (CaCt), LA, D, the daidzein combination D–CaCt–LA (D1:1:1), and the tempeh combination T–CaCt–LA (T1:1:1) was assessed through digestion simulation. The calcium content of both untreated and digested samples was determined using atomic absorption spectrometry (AAS). In the subsequent stage, the digested samples were used to induce intestinal absorption in differentiated enterocyte-like Caco-2 cells. The permeable fractions were then evaluated in a culture of osteoblast-like Saos-2 cells. Preliminary cellular experiments employed the MTT assay to assess cytotoxicity. The results indicated that the analyzed products did not influence the deposition of extracellular calcium in Saos-2 cells cultured without mineralization stimulators. The combined formulations of permeable fractions of digested CaCt, LA, D, and T demonstrated the capacity to enhance the proliferation of Saos-2 cells. In Saos-2 cells, D, D1:1:1, and LA showed no discernible impact on intracellular calcium accumulation, whereas T and T1:1:1 reduced the calcium deposits. Additionally, mRNA transcripts and alkaline phosphatase (ALP) activity levels in Saos-2 cells cultured without mineralization induction were unaffected by the analyzed products. An examination of the products revealed no discernible effect on ALP activity or mRNA expression during Saos-2 cell differentiation. Our findings suggest that tempeh, daidzein, and *L. acidophilus* did not positively impact cellular calcium deposition in Saos-2 cells. However, tempeh, daidzein and its combination, and *L. acidophilus* might enhance the process of osteogenic differentiation in Saos-2 cells. Nevertheless, this study did not identify any synergistic impact on calcium deposition and the process of osteogenic differentiation in Saos-2 cells of isoflavones and probiotics.

## 1. Introduction

Insufficient calcium intake significantly contributes to the onset of osteoporosis and an increased vulnerability to fractures [1]. The likelihood of calcium deficiency is notably pronounced in postmenopausal and elderly women, owing to age-related physiological and metabolic changes, along with heightened calcium requirements. These age-associated alterations frequently lead to inadequate calcium intake, compromised absorption, and altered calcium metabolism, collectively raising the risk of calcium deficiency. The interaction between calcium supply and its bioavailability plays a pivotal role in establishing optimal calcium levels and, consequently, in maintaining bone health [2].

The formation of bone tissue involves two distinct processes: endochondral ossification, characterized by the sequential development of a cartilage template followed by its replacement with bone tissue, and intramembranous ossification, where bone tissue forms directly through the concentration and osteogenic differentiation of mesenchymal stromal cells [3,4,5]. In intramembranous ossification, osteoblasts derived from mesenchymal stromal cells contribute to bone matrix deposition by generating collagen type I fibrils and regulating the deposition of minerals within the collagenous matrix [3]. Osteoblasts play a crucial role in the mineralization of the collagenous matrix, expressing proteins such as alkaline phosphatase (ALP), which provides the necessary phosphate for the mineralization process [6,7].

Recommendations underscore the importance of increased calcium intake as a preventive and therapeutic measure against bone loss, recognizing the pivotal role of calcium in determining bone health [8,9]. Calcium citrate demonstrates distinctive characteristics in absorption, solubility, bioavailability, tolerability, and compatibility with other substances, depending on factors such as a full stomach, fasting, or reduced gastric secretions. Administering calcium citrate between meals alleviates the competition with other nutrients, reduces the risk of renal calculus formation, and prevents abdominal distension and flatulence caused by carbon dioxide production. The suitability of calcium citrate for single-dose administration enhances the therapy flexibility and improves the potential for therapeutic compliance [10].

Probiotics and isoflavones have the potential to enhance calcium absorption and influence bone metabolism. Isoflavones are known to mobilize calcium from skeletal muscles, while probiotics impact calcium absorption through paracellular and transcellular mechanisms. *Lactobacillus* and *Bifidobacterium* nourish the gut microbiome, contributing to enhanced immunity and improved bone health. The microbial impact fosters the growth of beneficial microorganisms while inhibiting that of harmful ones. Through the promotion of osteoblast activity, limitation of bone resorption, and control of bone remodeling, isoflavones and their metabolites contribute to the improvement of bone mineral density [11].

Furthermore, research indicated that *Lactobacillus acidophilus* possesses osteoprotective properties that contribute to bone health. Ovariectomized mice administered *L. acidophilus* demonstrated improved bone microarchitecture, mineral density, and heterogeneity [12]. Daidzein is present in various soy products, especially in fermented soy like tempeh, where the concentration of isoflavones is notably higher compared to that in unfermented soy [13]. The gut microbiota metabolizes daidzein through fermentation, leading to the production of equol [14]. Equol provides protection against the development of osteoporosis in mice subjected to ovariectomy [15].

Intestinal cell culture models, such as Caco-2 cell lines derived from human colon adenocarcinoma, provide a valuable platform for enterocytic differentiation. These cells demonstrate increased levels of lactase, sucrase, and ALP, affirming their credibility as model systems. Morphological differentiation is marked by distinctive brush border membranes, and the existence of tight junctions signifies cellular polarization. Caco-2 cell monolayers function as effective models for studying the calcium paracellular transport pathway [16].

On the flip side, osteosarcoma, constituting approximately 20% of primary bone sarcomas, stands as the most prevalent malignant bone tumor. Various conventional subtypes, namely, osteoblastic, chondroblastic, and fibroblastic, alongside those with nonconventional morphologies like the telangiectatic and small cell subtypes, display variations in their predominant histologic features [17]. Saos-2 (sarcoma osteogenic), a non-transformed cell line derived from primary osteosarcoma cells with the ability to differentiate, has been employed in experimental studies due to its distinctive osteoblastic characteristics [18,19].

Numerous prior investigations scrutinized the bioavailability of calcium using in vitro cell models [20,21,22,23]. Concerns regarding the beneficial impacts of probiotics and isoflavones on calcium bioavailability have emerged as a rapidly expanding area of study. The inclusion of Saos-2 cells, as an osteoblast-like model, complements the use of Caco-2 cells, which simulate the absorption phase in an enterocyte-like fashion. Our primary objective was to investigate the potential interplay between gastrointestinal digestion products and bone metabolism. While Caco-2 cells simulate the absorptive aspect of the gastrointestinal process, Saos-2 cells offer insights into any downstream effects on bone cells. The rationale for employing Saos-2 cells was grounded in our interest in exploring potential systemic effects arising from the interaction of digested compounds with bone health. By examining the effects of these compounds on osteoblast-like cells, we aim to uncover their implications for bone metabolism and calcium bioaccessibility. This dual-cell model approach facilitates the exploration of the broader physiological impact of digested compounds beyond the gastrointestinal system, contributing to our understanding of potential connections between gastrointestinal digestion, absorption, and bone health. Thus, the present research delved into the impact of probiotics, isoflavones, and tempeh on the bioavailability of calcium. This was achieved by simulating digestion and using a cell model that involved human enterocyte-like Caco-2 and osteoblast-like Saos-2 cells. This is the first study that we are aware of that combines digestion simulation with cell assessment of the effects of probiotics and isoflavone products. A novel aspect of this study involved examining the impact of probiotics and isoflavones on cellular calcium deposition to determine their potential health benefits.

## 2. Results

### 2.1. Calcium Bioaccessibility

Table 1 displays the quantitative analysis of calcium content, calcium release, and potential bioaccessibility of calcium derived from soybean, tempeh, and a probiotic. Tempeh’s calcium content was found to be 72% higher than that of soybean, with tempeh also exhibiting a calcium release 2.5 times greater than soybean. Consequently, the percentage of potential calcium bioaccessibility for tempeh was approximately 110% higher compared to that for soybean. Despite having a lower calcium concentration compared to soybean products, the probiotic showed the highest calcium release and potential calcium bioaccessibility.

Table 2 presents the results of calcium release and potential bioaccessibility from two different sources: a combination of pure isoflavones (daidzein, calcium citrate, and probiotic) and a combination of tempeh (tempeh, calcium citrate, and probiotic). The combination of pure daidzein demonstrated a notable increase in both calcium release and calcium bioaccessibility when compared with tempeh-based combinations. Particularly, the use of a 1:1:1 formulation, whether comprising pure daidzein or tempeh, resulted in substantially elevated levels of calcium release and calcium bioaccessibility.

### 2.2. Effect on Cellular Calcium Deposition

Figure 1 explores the impact of several products, i.e., calcium citrate (CaCt), the probiotic *Lactobacillus acidophilus* (LA), daidzein (D), tempeh (T), and their combinations (D1:1:1 and T1:1:1), on the process of calcium deposition in Saos-2 cells. Subfigures (A,B) specifically demonstrate the effects on extracellular calcium transport. Subfigures (C,D) illustrate the dynamics of intracellular calcium transport, providing insights into the effects of the various treatments. Subfigures (E,F) examine ALP activity, accentuating variations in enzymatic activity associated with each treatment. Additionally, subfigures (G,H) offer an examination of ALP mRNA expression, elucidating the regulatory aspects of transcription in differentiated Saos-2 cells. The results suggest that the analyzed products did not influence extracellular calcium deposition in Saos-2 cells cultured without mineralization stimulators (Figure 1A,B).

Intracellular calcium assessment revealed that only CaCt increased the calcium content in Saos-2 cells cultured without mineralization inducers. D, D1:1:1, and LA did not impact intracellular calcium accumulation, unlike T and T1:1:1, which reduced the calcium deposits in Saos-2 cells (Figure 1C,D). Additionally, the analyzed products did not affect mRNA transcripts and ALP activity levels in Saos-2 cells cultured without inducing mineralization (Figure 1E–H).

Figure 2, visually representing Saos-2 cell cultures stained with alizarin red, highlights differences in cell monolayer morphology based on the supplemented product. The images reveal that intestinally permeable fractions from digested CaCt, LA, D, T, and their combined formulations could enhance Saos-2 cell proliferation. When compared to the nontreated control Saos-2 cell culture, monolayers with increased cell density were observed after treating Saos-2 cells with all analyzed products. Extracellular calcium deposition was detected in Saos-2 cell cultures stimulated by reference mineralization inducers (Figure 2). The promoting effect of the analyzed products on Saos-2 cell proliferation and viability was also evident in the cytotoxicity MTT test (Figure 3).

Figure 3 illustrates the effects on Saos-2 cell proliferation observed in Figure 2. These effects were noted for Saos-2 cells, CaCt, LA, D, T, and the combined formulations D1:1:1 and T1:1:1, with concentrations ranging from 0.05 to 1 mg/mL. Remarkably, these concentrations were deemed noncytotoxic, indicating that CaCt, LA, D, T, D1:1:1, and T1:1:1 promoted the proliferation of Saos-2 cells. The results emphasize the ability of concentrations within the 0.05 to 1.0 mg/mL range to enhance the proliferation of Saos-2 cells.

### 2.3. Effect on the Osteogenic Differentiation Process

Figure 4 illustrates the impact of CaCt, LA, D, T, D1:1:1, and T1:1:1 on the process of osteogenic differentiation. This figure provides a detailed exploration of the effects of various components, including calcium citrate (CaCt), the probiotic *L. acidophilus* (LA), daidzein (D), tempeh (T), and their combinations (D1:1:1 and T1:1:1), on the osteogenic differentiation process in Saos-2 cells. Subfigures (A,B) showcase the influence on extracellular calcium transport, revealing a significant increase induced by CaCt compared to the control group. Subfigures (C,D) illustrate the intracellular calcium transport dynamics, offering insights into the impact of the different treatments. Subfigures (E,F) display ALP activity, highlighting the variations in enzymatic activity associated with each treatment. Additionally, subfigures (G,H) provide a closer look at ALP mRNA expression, shedding light on its transcriptional regulation in differentiated Saos-2 cells.

Figure 5 visually presents the influence of calcium citrate, tempeh, daidzein, probiotic, and their combinations on Saos-2 cell cultures treated with an osteogenesis induction mixture (OIM). CaCt, T, LA, D, and D1:1:1 significantly increased the intracellular calcium content. Lower ALP activity was observed in differentiated Saos-2 cells under treatment with CaCt and T1:1:1. However, the ALP mRNA expression analysis did not confirm this effect (Figure 4G,H). The other analyzed products did not impact ALP activity following Saos-2 cell differentiation. Similarly, their possible effects on ALP mRNA expression were not detected.

## 3. Discussion

In this investigation, we explored the impact of tempeh, isoflavones, and a probiotic on calcium uptake using a cellular model. The significantly higher uptake of calcium induced by the D1:1:1 combination suggests its potential as a calcium source promoting osteogenesis. The elevated calcium content observed in tempeh compared to soybean positions it as a potentially valuable calcium source [24]. However, despite this potential, the current study revealed a relatively low bioaccessibility of calcium from tempeh. Meanwhile, our study did not uncover a positive influence of tempeh, daidzein, and *L. acidophilus*, either individually or in combination, on calcium deposition or ALP activity—a marker of osteogenesis. The observed increase in Saos-2 cell proliferation was evident only when osteogenesis was induced for all nutritional factors analyzed.

Despite the positive impact of tempeh on calcium release and bioaccessibility, it is crucial to note that both pure tempeh and the combination of tempeh with the probiotic and calcium citrate led to a significant reduction in intracellular calcium content during cellular calcium deposition, as illustrated in Figure 1. This phenomenon can be explained by the presence of antinutrient compounds, which have the potential to affect calcium availability [25]. In addition to these nutritional components, soybeans harbor antinutritional factors such as phytic acid [26], oxalate [27], fiber [28], tannins [29], and saponin [30]. These antinutritional factors influence the bioavailability of micronutrients, such as calcium, iron, copper, and zinc [31], acting as chelating agents in the gastrointestinal tract [32]. Results of other studies indicated that isoflavones, including daidzein, exhibit chelation activity towards metal ions [33]. The chelation activity may explain the relatively low calcium bioaccessibility and low calcium intracellular content in Saos-2 cells in the presence of tempeh. The chelation activity of isoflavonoids is a notable aspect that requires further research in light of our findings. Moreover, it is worth noticing that our samples were subjected to enzymatic digestion and might contain more isoflavone aglycones [34]. Isoflavone aglycones are absorbed faster and in greater amounts than their glucosides and may be more effective. The enzymatic process may have influenced the chelating activity of isoflavones and the calcium uptake. Fermentation results in a significant decrease in antinutritional factors [35], and tempeh was the focus of this study. The reduced calcium deposition in Saos-2 cells treated with tempeh was likely influenced by fermentation-induced changes in tempeh’s composition. For instance, fermentation leads to a reduction in phytic acid content, a compound known to decrease the bioavailable calcium [36,37]. It confirms that phytic acid has the potential ability to exert an inhibitory effect on calcium absorption [38,39,40].

While no statistically significant differences were observed, the products containing isoflavones, i.e., tempeh and pure daidzein, demonstrated the capacity to enhance ALP activity (Figure 1E,F) and ALP mRNA expression (Figure 1G,H) during cellular calcium deposition in Saos-2 cells. Our findings align with prior research, exemplified by studies indicating that soy isoflavones can elevate ALP activity after 7 and 14 days of treatment. Furthermore, in rat primary osteoblasts, treatment for 10 days resulted in increased mineralized nodule formation and calcium content within the mineralized nodules [41]. Moreover, isoflavone-enriched whole soymilk powder exhibited the ability to induce significant ALP activity in osteoprogenitors, even at an early treatment time of 48 h [42].

The observed lack of a significant improvement in ALP activity in response to daidzein, tempeh, and the probiotic in our cell study necessitates a careful consideration of the experimental context. The isolated cellular environment may not fully replicate the intricate interactions present in a living organism [43]. Notably, the absence of microbiota and isoflavone metabolites, such as equol, in our cell cultures may have contributed to the observed results, as these factors play a vital role in influencing ALP activity [44,45]. Additionally, the simplified conditions of cell studies may not fully capture the systemic situation present in vivo, including the interplay of endogenous factors [46], hormonal regulation [47], and growth factors [48], which collectively influence ALP activity. Moreover, variations in the concentration and exposure duration of the studied compounds could impact the outcome [49,50].

Our current study revealed that tempeh, a probiotic, and calcium citrate resulted in an increase in calcium intracellular deposition during the osteogenic differentiation process in Saos-2 cells (Figure 4). The observed effects might be linked to the formation of calcium phosphate facilitated by ALP activity in the medium. Moreover, the difference between ALP mRNA expression and ALP activity prompts a deeper investigation. This difference could stem from several biological factors, such as post-transcriptional modifications, translational regulation, or temporal variations in gene expression dynamics. This observed inconsistency underscores the complex and multifaceted nature inherent in cellular processes. In bone and calcifying cartilage, ALP is expressed early in development and is localized on the cell surface and within matrix vesicles. As the mineralized tissue matures, both ALP expression and activity typically decrease [6,7,51].

In the assessment of the effects of daidzein, calcium citrate, and *L. acidophilus* within the three-stage experimental model (Digestion-Caco2-Saos2), a thorough examination of potential interactions is crucial to comprehend the observed outcomes. Interestingly, our current study established that *L. acidophilus* did not manifest a synergistic effect with isoflavone products regarding cellular calcium deposition (Figure 1) and the osteogenic differentiation process (Figure 4). The dynamic interplay among these components is pivotal, as their collective impact on cellular processes may be influenced by antagonistic relationships. A conceivable antagonistic effect of probiotics involves a competition for nutrients and binding sites [52,53]. This competitive interaction introduces a layer of complexity that could contribute to the effects observed in the experimental setting.

As a result, a synergistic effect between isoflavones and probiotics in bone cellular metabolism was not evident in our current investigation. This outcome aligns with our earlier findings in healthy female rats [54,55,56]. This phenomenon can be attributed to the specific probiotic strain. For instance, in an in vitro study, Raveschot et al. [57] evaluated 174 *Lactobacillus* strains from Mongolian dairy products for their probiotic properties and impact on intestinal calcium absorption. Among the strains, *L. casei* 9b, *L. kefiranofaciens* 15b, *L. plantarum* 46a, *L. helveticus* 49d, and *L. delbrueckii* 50b displayed probiotic characteristics, influencing calcium transport in Caco-2 cells. Notably, *L. casei* 9b, *L. kefiranofaciens* 15b, and *L. helveticus* 49d enhanced the total calcium transport, likely through improved calcium solubility, while *L. delbrueckii* 50b impacted the paracellular pathway. *L. plantarum* 46a improved the calcium uptake via the transcellular pathway involving vitamin D receptor (VDR) and transient receptor potential cation channel subfamily V member 6 (TRPV6). Meanwhile, in an animal study by Scholz-Ahrens et al. [58], the combined supplementation of a specific prebiotic (oligofructose + acacia gum) with *L. acidophilus* NCC90 significantly prevented bone mineral loss in ovariectomized rats, highlighting the potential of synbiotics for maintaining bone health. Moreover, in a prior study involving postmenopausal women, the inclusion of *Bifidobacterium animalis* DN-173010 in a diet rich in isoflavones for an 8-week period demonstrated the capability to enhance parameters related to bone turnover during early menopause [59].

Moreover, the evaluated substances—specifically, tempeh, daidzein, *L. acidophilus*, and calcium citrate—were determined to be noncytotoxic (Figure 3). These results underscore the capacity of concentrations ranging from 0.05 to 1.0 mg/mL to enhance the proliferation of Saos-2 cells. Other studies support the nontoxic nature of daidzein when administered in high doses during experimental investigations [60,61]. Daidzein at a dose exceeding 5000 mg/kg did not exhibit adverse effects in a study of acute oral toxicity and, in a 28-day study with repeated oral doses of 25, 50, and 100 mg/kg, did not induce changes in hematology parameters, clinical biochemistry, or kidney function parameters [61]. The findings in this study lay the groundwork for further exploration in both experimental and clinical investigations.

A limitation of this study is that the in vitro digestion treatment utilized only pepsin and pancreatic enzymes. While this method represents a simplified digestive process, it has been employed in numerous studies [62,63]. Recent standardized in vitro digestion methods aim to simulate the physicochemical processes occurring in the human gastrointestinal tract (mouth, stomach, and small intestine) during food digestion [64,65,66,67]. Additionally, our study did not assess the expression of other metabolites. For instance, osteoblast differentiation requires the expression of bone morphogenetic protein 2 (BMP2), a crucial cytokine in bone formation and regeneration. Tartrate-resistant acid phosphatase (TRAP), a modulator of bone resorption, expressed at low levels in Saos-2 cells, is associated with conditions such as osteoporosis, osteoclastoma, and metabolic bone diseases when its expression is elevated [18].

## 4. Materials and Methods

### 4.1. Materials

Soybeans of the Augusta variety were acquired from the Department of Genetics and Plant Breeding at Poznań University of Life Sciences, Poland. *Rhizopus oligosporus* NRRL 2710 was obtained from the Agricultural Research Service Culture Collection (Peoria, IL, USA). Potato dextrose agar (PDA), skim milk, and maltodextrin were procured from Merck, Darmstadt, Germany. *Lactobacillus acidophilus* DSM20079 was sourced from the Leibniz-Institut Deutsche Sammlung von Mikroorganismen und Zellkulturen (DSMZ; German Collection of Microorganisms and Cell Cultures). De Man, Rogosa, and Sharpe (MRS) broth were obtained from Oxoid (Hampshire, UK). Porcine gastric mucosa-derived pepsin enzyme, porcine pancreas-derived pancreatin enzyme, hydrochloric acid (HCl), sodium bicarbonate (NaHCO_3_), and lanthanum chloride (LaCl_3_) were purchased from Sigma-Aldrich (Steinheim, Germany). Additionally, calcium citrate tetrahydrate was acquired from Warchem Sp. z o.o., (Warsaw, Poland). The materials used for cell assessments are thoroughly explained and outlined in the method descriptions, with all remaining chemicals adhering to analytical grade standards.

### 4.2. Tempeh Preparation

Tempeh was prepared following a methodology derived from our previous research [63], with certain adjustments for optimization. In summary, soybeans were dehulled before boiling for 40 min, followed by cooling. An inoculation period of 72 h was then implemented in PDA medium, during which *Rhizopus oligosporus* NRRL 2710 was introduced in the soybean preparation in disposable Petri dishes (15 cm in diameter). The fermentation proceeded for precisely 24 ± 1 h at a controlled temperature of 37 ± 1 °C. After completing the fermentation phase, the tempeh samples were frozen, subjected to freeze-drying, and subsequently processed into a powdered form.

### 4.3. Probiotic Preparation

*Lactobacillus acidophilus* DSM20079 was revived from a freeze-dried stock by inoculation into De Man, Rogosa, and Sharpe (MRS) broth, followed by a 1 h incubation at room temperature. The resulting suspension was spread onto MRS agar and cultivated at 37 °C for 24 h. A single colony from this culture was then subcultured in 10 mL of fresh MRS broth and incubated for 18 h at 37 °C. Subsequently, the culture volume was increased, and a portion was transferred into fresh MRS broth, obtaining a 2% culture using the inoculum from the overnight culture. The culture was later centrifuged at 4500× *g* for 10 min at 4 °C to harvest the cells, which were washed once with cold sterilized distilled water. The cells were resuspended at 10^11^ CFU/mL in a mixture containing 10% skim milk powder and 20% maltodextrin, both cold-sterilized.

The cell mixture was poured into sterile plates, frozen at −80 °C, and subsequently freeze-dried at room temperature with the condenser temperature set at 55 °C. Random samples of freeze-dried material were taken, and 1 g of each sample was suspended in 9 mL of 0.85% normal saline. These suspensions were manually homogenized under aseptic conditions to enumerate the viable cells. A 1 mL aliquot was taken from each suspension, and serial dilutions were prepared in 0.85% sterile saline. The appropriate dilutions were then pour-plated onto sterile MRS agar and anaerobically incubated at 37 °C. The resulting colonies were counted. The final formulation of the probiotic preparation incorporated corn starch and maintained a concentration of 10^10^ CFU/gram.

### 4.4. Experimental Design

To accomplish the objectives of this study, we implemented a comprehensive methodology comprising two distinct steps. The visual representation of the research design workflow is illustrated in Figure 6. In the initial step, we commenced the investigation by performing digestion simulations to evaluate the bioaccessibility of calcium sourced from three entities: calcium citrate, a probiotic, daidzein, and their combinations. Following this, the calcium content in both the native and the digested samples was quantified using atomic absorption spectrometry (AAS).

In the second phase, the gastrointestinally digested samples were administered to differentiated enterocyte-like Caco-2 cells to simulate intestinal absorption. Following this, we examined the fractions that had undergone intestinal permeation in the culture of osteoblast-like Saos-2 cells. In the initial cell experiments, we conducted cytotoxicity assessments using the MTT assay. This segment of our study aimed to evaluate the impact of the selected samples within the simulated intestinal environment provided by the Caco-2 epithelium and to assess their effects on Saos-2 cells, particularly concerning bone mineralization processes. The analyzed samples in this study were formulated individually and in combination. The specific formulations for the individual and combined samples are detailed in Table 3.

### 4.5. Step 1: Digestion Simulation

The in vitro digestion study was designed to evaluate the bioaccessibility of calcium from various sources, including calcium citrate, probiotics, daidzein, tempeh, and their combined formulations. The digestion simulation followed a modified version of our established methodology. Briefly, the samples under investigation were placed in conical flasks, and 20 mL of deionized water was added to each flask. The samples underwent 10 min of homogenization. The digestion procedure included the following steps. First, the pH of the samples was adjusted to 2 using a 0.1 M HCl aqueous solution to activate the pepsin enzyme. Subsequently, a pepsin solution (0.5 mL/100 mL) was added. The samples were incubated in a thermostat-controlled shaker (Benchmark Scientific, Sayreville, NJ, USA) at 37 °C for 2 h, with pH adjustments using a 6 M HCl aqueous solution as needed during this incubation phase. Second, after the initial 2 h incubation, the digested samples underwent pH adjustment to 6.8–7.0 with a 6% NaHCO_3_ aqueous solution, followed by the addition of a pancreatin solution (10 mL/40 mL homogenate). The samples were then placed back into the thermostat-controlled shaker at 37 °C for 4 h. Third, the digested samples were carefully transferred into conical centrifuge tubes (MPW Med. Instruments, Warsaw, Poland). Finally, the samples were centrifuged at 3800 rpm for 10 min. The calcium concentration of each sample was determined by comparison to a blank sample consisting of deionized water and reagents.

### 4.6. Determination of Calcium Content in Native Samples and Digested Samples

The quantification of calcium content in both the native and the in vitro digested samples was conducted using atomic absorption spectrometry (AAS-3, Carl Zeiss, Jena, Germany). To determine the total calcium content in the native samples, a dry mineralization process was employed. Specifically, each powdered sample, totaling 2 g, underwent ashing in a muffle furnace set at 450 °C, continuing until complete mineralization was achieved. The resulting ashes were then solubilized in 1 N nitric acid (Suprapure, Merck KGaA, Darmstadt, Germany).

Simultaneously, wet mineralization was performed by introducing 65% nitric acid into the digested samples, which were then subjected to mineralization within a Speedwave XPERT Microwave Digestion System (Berghof, Eningen, Germany). The spectrometer was configured to the calcium wavelength of 422.7 nm. The accuracy of the analytical method for calcium determination was established to be 92% and further validated through the simultaneous analysis of reference material (soya bean flour, INCT-SBF-4, LGC standards, Teddington, UK). The calcium content of the samples is expressed as mg/100 g, while calcium bioaccessibility is indicated as a percentage of the total content released.

### 4.7. Step 2: Intestinal Transport

#### 4.7.1. Intestinal Caco-2 Epithelium Model

The human intestinal epithelial Caco-2 cell line (HTB-37^™^) was procured from the American Type Culture Collection (ATCC, Manassas, VA, USA). The cells were cultivated in Dulbecco’s Modified Eagle Medium (DMEM, Sigma-Aldrich), supplemented with 1% nonessential amino acids (100× NEAA, Sigma-Aldrich), 20% fetal bovine serum (FBS, Gibco BRL, Grand Island, NY, USA), and gentamicin (50 mg/L), and maintained at 37 °C in a humidified atmosphere of 95% air and 5% CO_2_.

For the formation of the intestinal barrier, Caco-2 cells were seeded on polycarbonate membranes with a pore size of 0.4 μm (3.14 cm^2^) (Nunc^TM^ polycarbonate cell culture inserts) at an initial density of 4 × 10^5^ cells/cm^2^. The cells were cultured for 21–22 days, changing the medium three times a week. The integrity of the Caco-2 cell monolayers was assessed through transepithelial electrical resistance (TEER) measurements using the Millicell Electrical Resistance System (ERS-2, MilliporeSigma, Burlington, MA, USA). Caco-2 cell cultures with TEER values ≥ 600 Ω × cm^2^ were utilized in the transport experiments.

#### 4.7.2. Cytotoxicity Analysis

Caco-2 cells were initially seeded at a density of 2 × 10^4^ cells/cm^2^ and cultured under standard conditions. After 24 h, the cells were treated for 48 h with digested calcium citrate, probiotic, daidzein, and their combinations at concentrations ranging from 0 to 1 mg/mL. Cytotoxicity was assessed using the 3-(4,5-dimethylthiazol-2-yl)-2,5-diphenyl-2H-tetrazolium bromide (MTT) assay. In essence, the MTT reagent was introduced into the Caco-2 cell culture to achieve a final concentration of 0.5 mg/mL. Following a 3 h incubation, formazan crystals were extracted from the cells using acidic isopropanol. Absorbance was measured using a Tecan Infinite M200 microplate reader (Tecan Group Ltd., Männedorf, Switzerland) at the wavelengths of 570 nm and 690 nm.

#### 4.7.3. Sample Preparation and Transport Experiment

Figure 7 illustrates the cell experiments conducted on human intestinal epithelial Caco-2 cells and osteoblast-like Saos-2 cells. The samples were dispersed in a transport medium (HBSS buffer w/o Ca and Mg, pH 7.2–7.4) at a maximum concentration of 5 mg/mL, determined to be noncytotoxic to Caco-2 cells based on preliminary cytotoxicity tests. The suspensions were sterilized through filtration using 0.22 μm pore-size membranes. The sterile samples were introduced onto the apical (donor) side, representing the intestinal lumen, of the two-compartment intestinal barrier Caco-2 model. Saos-2 cell culture medium (McCoy’s medium with 15% FBS) was placed on the basolateral (acceptor) side. The calcium extracellular content explicitly represents the average amount of calcium that has successfully traversed the cellular barrier, specifically referring to the quantity of calcium present in the basolateral compartment.

Intestinal transport was carried out at 37 °C under shaking (100 rpm). Following a 2 h duration, the sample fraction transported across the Caco-2 barrier was analyzed in Saos-2 cell cultures. The TEER parameter was monitored both before and after the transport experiment to ensure the integrity of the intestinal barrier.

### 4.8. Osteogenic Experiments Using Saos-2 Cells

#### 4.8.1. Osteoblast-like Saos-2 Cell Culture

Saos-2 cells (HTB-85), from a human osteosarcoma cell line, were cultured in McCoy’s medium formulated by the ATCC, supplemented with 15% FBS and gentamicin (50 mg/L), and maintained at 37 °C in a humidified atmosphere of 95% air and 5% CO_2_. The impact of the intestinally permeable fractions of the digested samples on cellular calcium deposition was assessed in Saos-2 cells cultured under standard conditions (without osteogenesis inducers) or in osteogenesis induction medium (OIM), following established procedures.

Saos-2 cells were initially seeded at a density of 4 × 10^4^ cells/cm^2^ and cultured until confluence, with medium replacement every 48 h. Upon reaching confluency, the osteogenesis process was initiated using the osteogenesis induction medium (OIM), consisting of McCoy’s medium, 15% FBS, 10 mM β-glycerophosphate, 100 nM dexamethasone, and 50 μg/mL of L-ascorbic acid. OIM was refreshed every 3 days, and the cells were cultured for a duration of 15 days. Each time the culture medium was changed, the analyzed intestinally permeabilized fractions were administered to Saos-2 cells growing in medium without or with differentiation inducers. Following the 15-day culture period, cellular calcium accumulation was quantified, encompassing extracellular calcium deposition, determined through the alizarin red assay, and intracellular calcium content, obtained via a calcium colorimetric assay. Additionally, the activity and mRNA expression of ALP, an early osteogenic marker, were determined using an ALP assay kit and real-time PCR. All experiments and analyses were conducted in triplicate, and the presented results are expressed as means with standard deviation (±SD).

#### 4.8.2. Alizarin Red Staining and Quantification Assay

Following the 15-day culture period, the cells underwent PBS (phosphate-buffered saline) washing and fixation with 10% formaldehyde at room temperature for 15 min. Subsequently, the cells were rinsed thrice with ddH_2_O and stained with a 40 mM alizarin red solution (Sigma-Aldrich, Merck Group) at room temperature for 30 min. After staining, the cells were washed four times with ddH_2_O. For quantification, the stained mineralized nodules were subjected to a 30 min incubation with 10% cetylpyridinium chloride (Sigma-Aldrich). The resulting solutions were collected, and absorbance at 562 nm was measured using a Tecan Infinite M200 microplate reader. Sample quantification was performed based on the alizarin red standard curve. The inclusion of alizarin red staining, a widely accepted method for assessing calcium deposition, complemented the quantitative measures by providing a qualitative and visual evaluation of mineralization [68].

#### 4.8.3. Intracellular Calcium Assay

After a 15-day culture, the cells were washed with PBS and lysed with RIPA (radio-immunoprecipitation assay) lysis buffer at room temperature for 30 min, and the lysates were combined with an equal volume of 1 M HCl. The RIPA lysis buffer consists of a detergent, salts, and protease inhibitors, facilitating the efficient extraction of proteins from cells for subsequent analysis. This mixture was then incubated at 4 °C overnight.

The calcium content in the lysates was determined using a colorimetric calcium assay kit (Sigma-Aldrich) suitable for calcium measurements in tissue homogenates and cell lysates. The calcium ion concentration was determined by quantifying the chromogenic complex formed between calcium ions and o-cresolphthalein, which was measured at 575 nm (Tecan Infinite M200 microplate reader). The assay was performed according to the protocol recommended by the manufacturer.

#### 4.8.4. Alkaline Phosphatase Activity Assay

The quantitative assessment of ALP activity in Saos-2 cells was conducted using an ALP assay kit (Sigma-Aldrich), following the manufacturer’s instructions. Post treatment, the cells were washed with PBS and then lysed in a 0.2% Triton X-100 solution at room temperature for 20 min. For the ALP activity measurements, *p*-nitrophenyl phosphate was employed as the substrate. This substrate undergoes hydrolysis by ALP, resulting in a yellow-colored product with a maximum absorbance at 405 nm. Since this assay is based on a kinetic reaction, absorbance was measured immediately (*T* = 0 min) and again after 4 min (*T* = 4 min) using a Tecan Infinite M200 microplate reader. The obtained data were normalized to the cellular protein content, with total protein quantification carried out through the BCA assay (Pierce^®^ BCA Protein Assay Kit, Thermo Scientific Inc., Waltham, MA, USA) following the manufacturer’s protocol.

#### 4.8.5. Alkaline Phosphatase mRNA Expression Analysis

Gene expression analysis followed a previously outlined protocol [69]. The TRI reagent (Sigma-Aldrich) facilitated total RNA isolation, the cDNA Transcriptor First-Strand kit (Roche Diagnostics GmbH, Mannheim, Germany) was employed for initial cDNA synthesis, and SYBR^®^ Select Master Mix (Life Technologies, Carlsbad, CA, USA) was used for real-time PCR. The primers for cDNA amplification were as follows: forward, 5′-GACCCTTGACCCCCACAAT-3′ and reverse, 5′-GCTCGTACTGCATGTCCCCT-3′ (product size 68). The transcript levels were normalized using glyceraldehyde 3-phosphate dehydrogenase (GAPDH) as an internal control, with the following primer sequences: forward, 5′-TGCACCACCAACTGCTTAGC-3′ and reverse, 5′-GGCATGGACTGTGGTCATGAG-3′ (product size 87). The relative mRNA expression is presented as fold change, calculated using the 2^−ΔΔCt^ method, compared to control cells.

### 4.9. Statistical Analysis

The normality of the data distributions was evaluated using the Shapiro–Wilk test. Following this, analysis of variance (ANOVA) was employed to ascertain statistical significance, with subsequent Tukey’s post hoc honest significant difference test. For data with non-Gaussian distribution, we used the Kruskal–Wallis test. All identified differences reached statistical significance at a significance level of 5%. The statistical analysis was executed using SPSS version 22 on the Windows operating system.

## 5. Conclusions

In summary, this study aimed to assess the impact of tempeh, pure daidzein, and *Lactobacillus acidophilus* on calcium uptake and deposition in Saos-2 cells, with which the bone mineralization process was simulated. In the initial phase, we evaluated calcium bioaccessibility from these nutrients and their combinations through digestion in an artificial gastrointestinal tract. Subsequently, the digested products were subjected to simulated intestinal absorption in the intestinal epithelial Caco-2 cell model, and the intestinal permeable fractions were scrutinized in a culture of osteoblast-like Saos-2 cells. This methodology allowed for exploring the interactions and effects of the studied products on calcium bioavailability and bone-related cellular processes in our experimental models.

Our findings suggest that daidzein, tempeh, and *L. acidophilus* do not have a beneficial effect on cellular calcium deposition in Saos-2 cells. However, tempeh, daidzein, and their combinations, along with *L. acidophilus*, may enhance the osteogenic differentiation process in Saos-2 cells. Notably, no synergistic effect on calcium deposition and the osteogenic differentiation process in Saos-2 cells between the studied isoflavones and probiotic was observed in this study.

Future investigations should delve into the intricate molecular pathways that underlie the observed effects of daidzein, tempeh, and *L. acidophilus* on the osteogenic differentiation process in Saos-2 cells. Long-term studies are essential to evaluate the sustained effects of these nutrients, considering factors such as prolonged exposure, cellular adaptation, and potential cumulative benefits. Additionally, employing in vivo models and clinical trials could validate and extrapolate to more biologically significant conditions these findings from cell culture, providing a more comprehensive understanding of their translational potential and of safety considerations for incorporating these elements into interventions aimed at preserving or enhancing bone health.

## Figures and Tables

**Figure 1 ijms-25-01008-f001:**
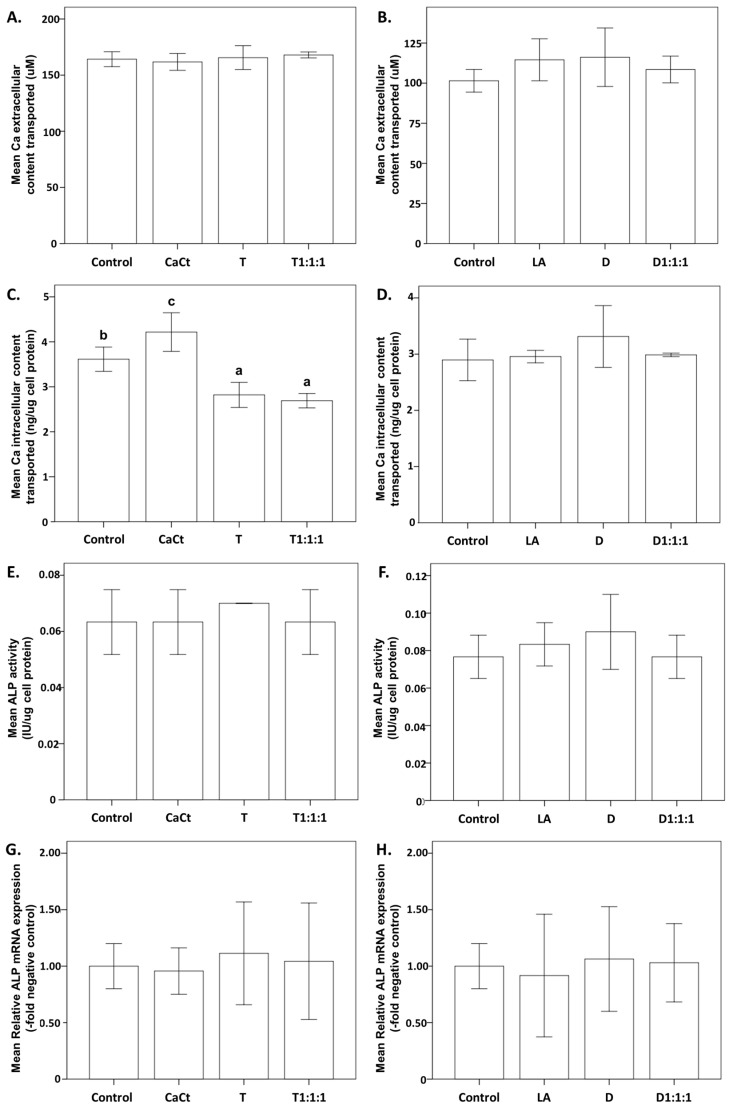
Impact of calcium citrate (CaCt), the probiotic *Lactobacillus acidophilus* (LA), daidzein (D), tempeh (T), and the combinations of tempeh, calcium citrate, and the probiotic *L. acidophilus* (T1:1:1) and of daidzein, calcium citrate, and the probiotic *L. acidophilus* (D1:1:1) on extracellular calcium deposition (**A**,**B**), intracellular calcium content (**C**,**D**), ALP activity (**E**,**F**), and ALP mRNA expression (**G**,**H**) in Saos-2 cells. The analyzed intestinally permeabilized fractions were introduced into Saos-2 cell cultures throughout the culture medium every 3 days for 15 days. Significant differences between means within each figure are denoted by distinct letters (a–c), and these letters represent comparisons with a significance level of *p* < 0.05. The values are expressed as means ± standard deviation. Each figure includes a control group with triplicate analyses. This figure exclusively displays statistically analyzed data, highlighting statistically significant differences.

**Figure 2 ijms-25-01008-f002:**
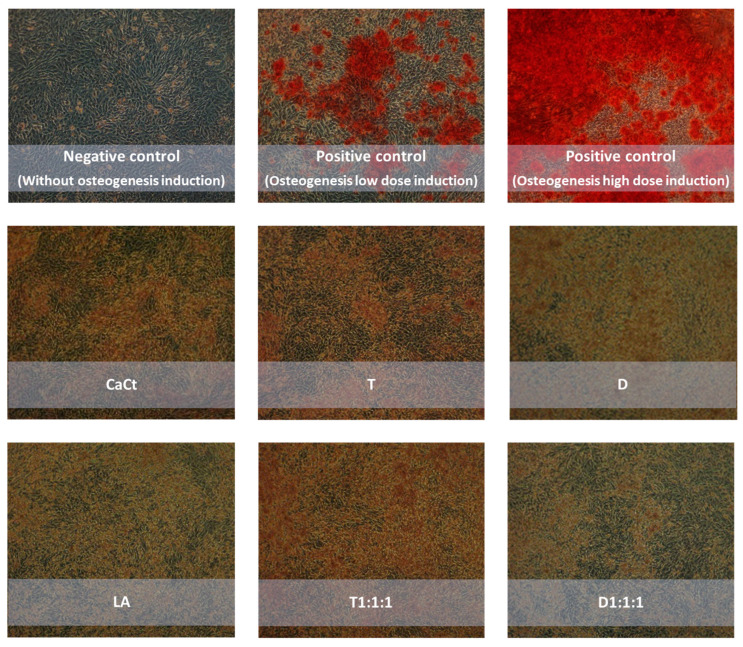
Impact of intestinally permeabilized fractions of calcium citrate (CaCt), tempeh (T), daidzein (D), the probiotic *Lactobacillus acidophilus* (LA), and the combinations of D, CaCt, and LA (D1:1:1) and of T, CaCt, and LA (T1:1:1) on extracellular calcium deposition in Saos-2 cell cultures following a 15-day treatment. In the negative control, cells were cultured without any analyzed fractions and osteogenic mixtures. In the positive controls, the Saos-2 cells were induced by osteogenic medium addition with low and high concentrations of osteogenesis inducers. Extracellular calcium deposits in the treated Saos-2 cells were determined by the alizarin red staining method. Photos were taken at 100× magnification.

**Figure 3 ijms-25-01008-f003:**
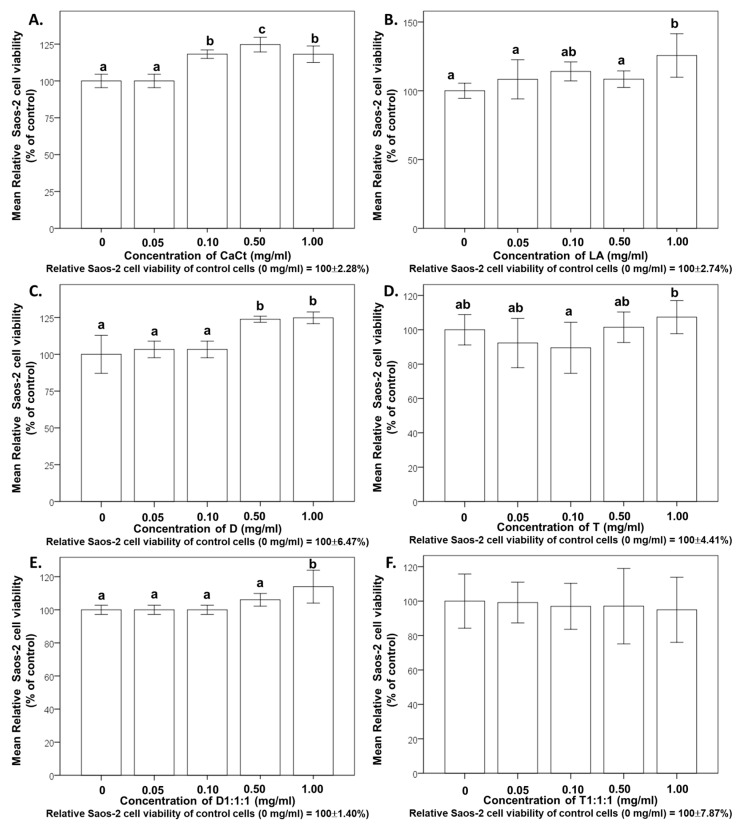
Impact of calcium citrate (CaCt) (**A**), the probiotic *Lactobacillus acidophilus* (LA) (**B**), daidzein (D) (**C**), tempeh (T) (**D**), and the combinations of D, CaCt, and LA (D1:1:1) (**E**) and of T, CaCt, and LA (T1:1:1) (**F**) on Saos-2 cell proliferation, viability, and metabolic activity determined using the MTT test. The cells were treated with CaCt, LA, D, T, T1:1:1, and D1:1:1 at concentrations ranging from 0.05 mg/mL to 10 mg/mL for 48 h. Significant differences between means within each figure are denoted by distinct letters (a,b or a–c), and these letters represent comparisons with a significance level of *p* < 0.05. The values are expressed as means ± standard deviation. Statistical analyses were conducted with triplicate measurements for each figure, resulting in different letter notations for significant differences. This figure exclusively displays statistically analyzed data, highlighting statistically significant differences.

**Figure 4 ijms-25-01008-f004:**
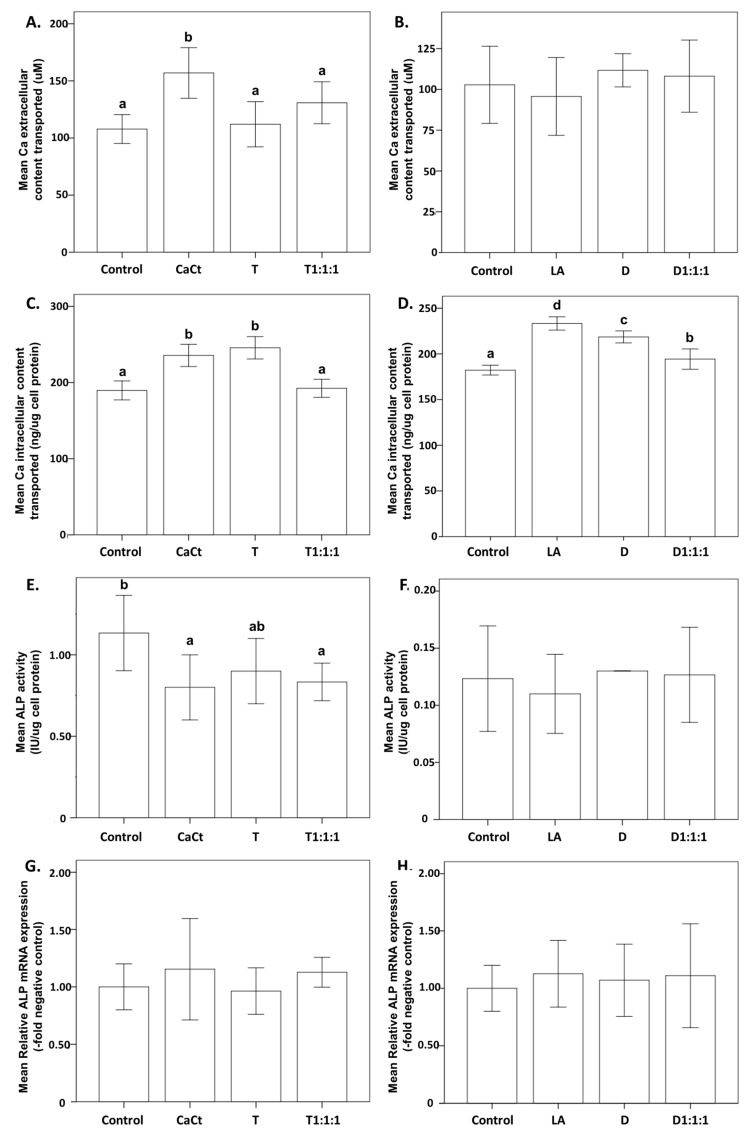
Impact of gastrointestinally digested and intestinally transported calcium citrate (CaCt), probiotic *Lactobacillus acidophilus* (LA), daidzein (D), tempeh (T), and combinations of T, CaCt, and LA (T1:1:1) and of D, CaCt, and LA (D1:1:1) on extracellular calcium deposition (**A**,**B**), intracellular calcium content (**C**,**D**), ALP activity (**E**,**F**), and ALP mRNA expression (**G**,**H**) in Saos-2 cells. The analyzed products were introduced into Saos-2 cell cultures during the osteogenesis process induced by the osteogenesis induction medium, which was refreshed every 3 days for 15 days. Significant differences between means within each figure are denoted by distinct letters (a,b or a–d), and these letters represent comparisons with a significance level of *p* < 0.05. The values are expressed as means ± standard deviation. Statistical analyses were conducted with triplicate measurements for each figure, resulting in different letter notations for significant differences. This figure exclusively displays statistically analyzed data, highlighting statistically significant differences.

**Figure 5 ijms-25-01008-f005:**
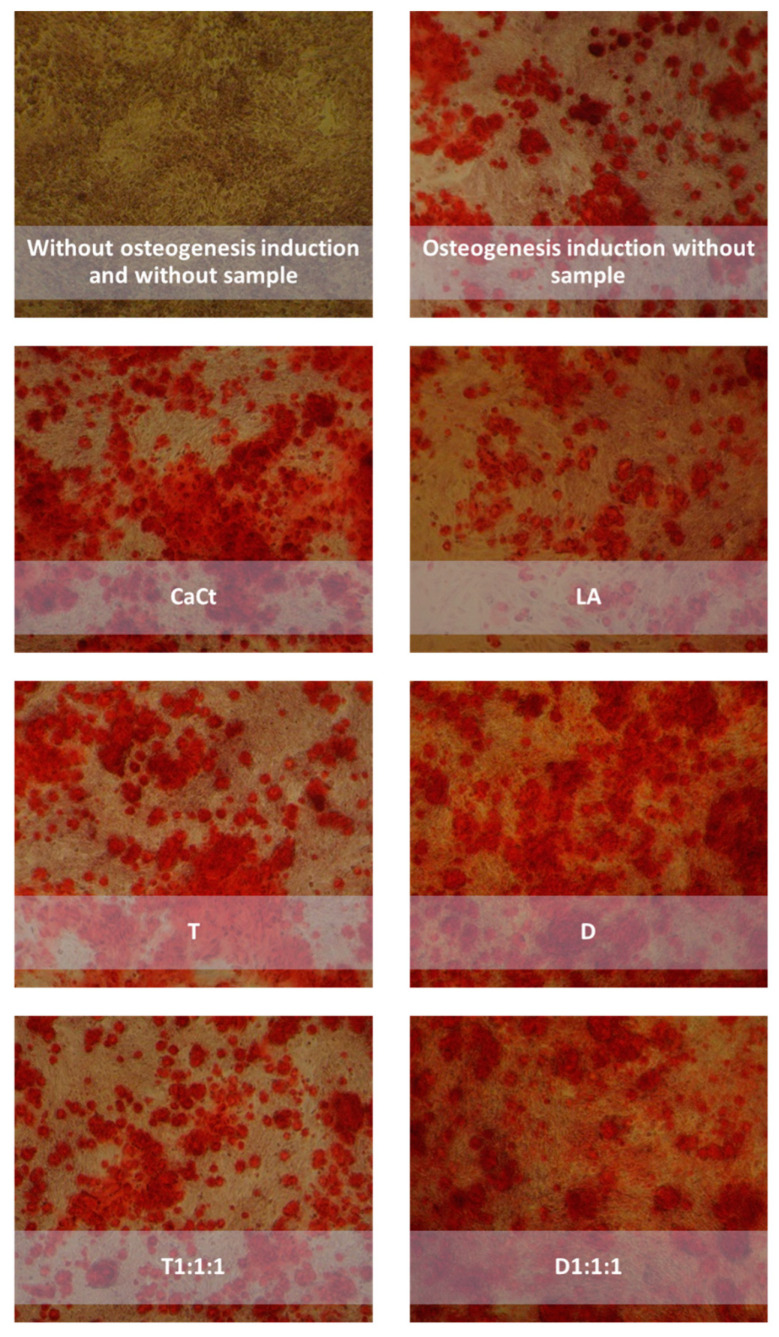
Impact of gastrointestinally digested and intestinally transported calcium citrate (CaCt), probiotic *Lactobacillus acidophilus* (LA), daidzein (D), tempeh (T), and tempeh (T1:1:1) or daidzein (D1:1:1) mixtures (combinations of tempeh or daidzein, calcium citrate, and probiotic *L. acidophilus*) on extracellular calcium deposition in Saos-2 cell cultures following a 15-day treatment with osteogenesis inducers. In the negative control, the cells were cultured without any analyzed fractions and osteogenic mixtures. In the positive controls, Saos-2 cells were induced by osteogenic medium addition. Extracellular calcium deposits in the treated Saos-2 cells were determined by the alizarin red staining method. Photos were taken at 100× magnification.

**Figure 6 ijms-25-01008-f006:**
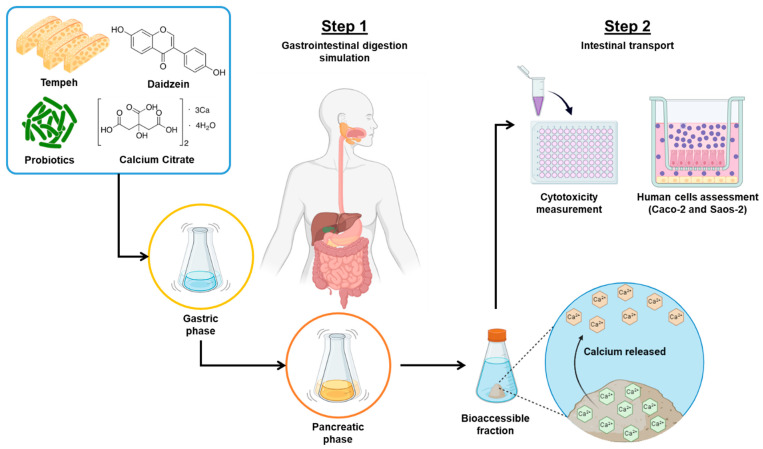
Experimental design of in vitro digestion with human cell assessments.

**Figure 7 ijms-25-01008-f007:**
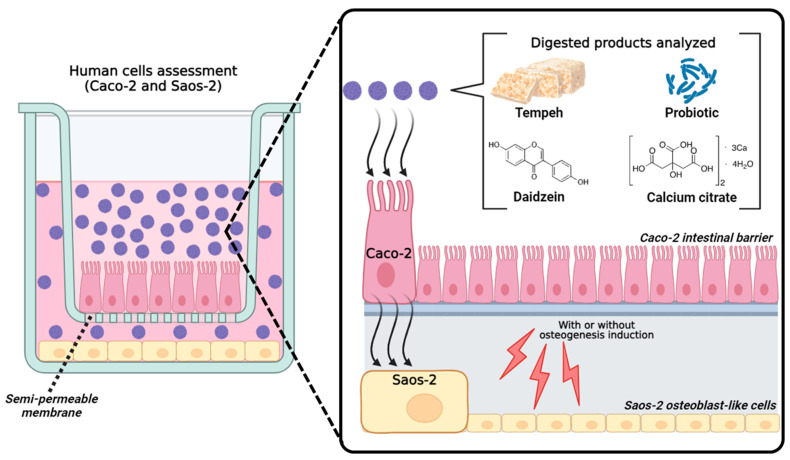
Illustration of the cell experiments performed using the human intestinal epithelial Caco-2 cells and the osteoblast-like Saos-2 cells.

**Table 1 ijms-25-01008-t001:** Calcium content in soybean, tempeh, and a probiotic.

Native Sample	Calcium Content (mg/100 g)	Calcium Release (mg/100 g)	Calcium Bioaccessibility (%)
Soybean	234.77 ± 15.83 ^b^	4.04 ± 0.04 ^a^	1.72 ± 0.02 ^a^
Tempeh	402.04 ± 11.31 ^c^	14.49 ± 2.57 ^b^	3.60 ± 0.64 ^b^
Probiotic	115.15 ± 8.02 ^a^	20.80 ± 0.02 ^c^	18.06 ± 0.02 ^c^

Significant differences between means within each column are denoted by distinct letters (^a–c^), and these letters represent comparisons with a significance level of *p* < 0.05. The values are expressed as means ± standard deviation. Calcium release is expressed as mg/100 g, representing the absolute amount of calcium released from the digested samples. Calcium bioaccessibility is expressed as a percentage, reflecting the relative availability of calcium for absorption.

**Table 2 ijms-25-01008-t002:** Calcium bioaccessibility in the digested combination samples.

Digested Sample	Calcium Release (mg/100 g)	Calcium Bioaccessibility (%)
D1:1:1	17.38 ± 0.37 ^b^	15.28 ± 0.33 ^b^
T1:1:1	6.12 ± 0.83 ^a^	1.19 ± 0.16 ^a^

Significant differences between means within each column are denoted by distinct letters (^a,b^), and these letters represent comparisons with a significance level of *p* < 0.05. The values are expressed as means ± standard deviation. Calcium release is expressed as mg/100 g, representing the absolute amount of calcium released from the digested samples. Calcium bioaccessibility is expressed as a percentage, reflecting the relative availability of calcium for absorption.

**Table 3 ijms-25-01008-t003:** Composition of the samples for calcium content and calcium bioaccessibility analyses.

Sample Code	Formula			
	Daidzein	Tempeh	Calcium Citrate	Probiotic
CaCt	-	-	1	-
LA	-	-	-	1
D	1	-	-	-
T	-	1	-	-
D1:1:1	1	-	1	1
T1:1:1	-	1	1	1

CaCt: calcium citrate; LA: probiotic *Lactobacillus acidophilus*; D: daidzein; T: tempeh. The amount of added calcium citrate was standardized to 1 mg of calcium. Daidzein and tempeh were included at doses corresponding to the recommended daily intake of isoflavones, set at 100 mg. The probiotic *L. acidophilus* was introduced at a dose aligning with the recommended daily probiotic intake range, falling between 10^8^ and 10^10^ CFU/g. The numerical value “1” represents a single quantity.

## Data Availability

The data presented in this study are available on request from the corresponding author.

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
