# Peer review of "Effects of Daidzein, Tempeh, and a Probiotic Digested in an Artificial Gastrointestinal Tract on Calcium Deposition in Human Osteoblast-like Saos-2 Cells"

_ijms, 2024, doi:10.3390/ijms25021008_

Round 1

Reviewer 1 Report

Comments and Suggestions for Authors

The paper of Harahap et al. is a potentially interesting paper. There are however some major issues which have to be explained before this paper will be acceptable for publication.

1.       The background of this research is not very clear, why to test digested probiotics, tempeh and soybeans on sarcoma cell lines Saos-2. I understand the testing with Caco-2 cells, as digested compounds are in contact with them but physiological or pathological relevance of the former testing is very unclear.

2.       The results part is not clearly written and the same is true for some methodological parts.

A.      Authors are using terms calcium release and calcium bioaccessibility, it is not clear, where is the difference. Moreover, authors tested incubation of digested probiotics, tempeh and soybeans and their combination on Caco-2 cells, where are the data on calcium penetration through these Caco-2 cells?

B.      Also other data are very unclear, what does it mean „Mean Ca extracellular content transported“ – the amount of calcium transported through the cells, i.e. in the basolateral compartment ? This must be clearly specified

C.      Authors should explain the use of Alizarin Red, why it was used if authors had atomic absorption spectrometry, which is more precise, at disposal

D.      In general, legend to the figures should be markedly improved, e.g. by adding some crucial methodological details such as time of incubation

3.     I am missing in some graphs negative control samples, e.g. authors are claiming at r. 208, that increase in Saos-2 cell proliferation was evident only when osteogenesis was induced for all nutritional factors analyzed. E.g., in Figure 3, the are no control cells.

Smaller issues:

The title should be modified as the expression „on calcium cellular bioavailability“ is not correct,  biovailability is a well-defined term a cellular bioavailability does not exist. similarly, see the phrase at r. 200 in the discussion

The use of designation A1,A2,B1 etc. in graphs is uncommon and confusing, the standard designation A,B,C etc. should be used instead

Discussion should be extended in two aspects

1.       The effect of pH and enzymes on the tested food and bioactives, e.g. Penha et al.. Enzymatic pretreatment in the extraction process of soybean to improve protein and isoflavone recovery and to favor aglycone formation. Food Res Int. 2020 Nov;137:109624

2.       Chelation activity of isoflavonoids - Daidzein tended to increase intracellular Ca content (Fig 1-B2). Isoflavonoid are known chelators, e.g.  Karlíckova et al. Isoflavones Reduce Copper with Minimal Impact on Iron In Vitro. Oxid Med Cell Longev. 2015;2015:437381  These data can extend the discussion in row 216 and later

Some parts are unclear – r. 247 „These effects may be attributed to the deposition of calcium phosphate resulting from ALP activity in the medium before the cells express ALP.“  Or r. 357 – „Subsequently, the intestinally permeable fractions were scrutinized in the osteoblast-like Saos-2 cell culture.“ or similar expression at r. 456.

r. 419 - at concentrations ranging from 010 mg/ml ?

r. 485 what is RIPA lysis buffer?

Para 4.8.3 – specify the mechanism of Colorimetric Calcium Assay kit

r. 516 – Normality of variable distributions  - normality of data distribution - When the normality of distribution was not found, which statistical test(s) was(were) used?

Subscripts and superscripts are sometimes missing (r. 313, 501, 507)

Comments on the Quality of English Language

Quite fine, see my detailed comments

Author Response

Response to Reviewer 1 Comments

Thank you very much for taking the time to review this manuscript. Please find the detailed responses below and the corrections highlighted in the re-submitted files.

Point-by-point response to Comments and Suggestions for Authors

The paper of Harahap et al. is a potentially interesting paper. There are however some major issues which have to be explained before this paper will be acceptable for publication.

Comments 1: The background of this research is not very clear, why to test digested probiotics, tempeh and soybeans on sarcoma cell lines Saos-2. I understand the testing with Caco-2 cells, as digested compounds are in contact with them but physiological or pathological relevance of the former testing is very unclear.

Response 1:

Thank you very much for your insightful comments on our manuscript. We appreciate the opportunity to address your concerns and enhance the clarity of our research.

1.     Background Clarification:

We appreciate the reviewer’s concern and understand the need for a clearer background. The rationale behind testing digested probiotics, tempeh, and soybeans on Saos-2 cells lies in exploring the potential interplay between gastrointestinal digestion products and bone metabolism. Saos-2 is a human osteosarcoma cell line that displays several osteoblastic features. These cells express receptors for 1,25-dihydroxyvitamin D3 and have high basal alkaline-phosphatase activity. They express the parathyroid hormone (PTH) receptor and produce cyclic AMP in response to treatment with PTH. These cells do not form tumors when injected subcutaneously into immunocompromised mice. When injected into diffusion chambers implanted intra-peritoneally into immunocompromised mice, however, Saos-2 cells produce a mineralized matrix, a defining characteristic of osteoblastic cells. These characteristics make this cell line an attractive source of bone-related molecules for research.

2.     Physiological or Pathological Relevance:

We acknowledge the importance of establishing physiological or pathological relevance. Our rationale for testing on Saos-2 cells is rooted in the potential systemic effects that digested compounds may have on bone health. By examining the interaction of these compounds with osteoblast-like cells, we aim to uncover any implications for bone metabolism and calcium homeostasis. This dual-cell model approach allows us to explore the broader physiological impact beyond the gastrointestinal system and delve into potential connections between gastrointestinal health and bone health.

We hope these clarifications address your concerns and contribute to the overall coherence of our manuscript.

Revision in the text (Lines 100-111):

The inclusion of Saos-2 cells, as an osteoblast-like model, complements the use of Caco-2 cells, which simulate the absorption phase in an enterocyte-like fashion. Our primary objective is to investigate the potential interplay between gastrointestinal digestion products and bone metabolism. While Caco-2 cells simulate the absorptive aspect, Saos-2 cells offer insights into any downstream effects on bone cells. The rationale for employing Saos-2 cells is grounded in our interest in exploring potential systemic effects arising from the interaction of digested compounds with bone health. By examining the effects of these compounds on osteoblast-like cells, we aim to uncover implications for bone metabolism and calcium bioaccessibility. This dual-cell model approach facilitates an exploration of the broader physiological impact beyond the gastrointestinal system, contributing to our understanding of potential connections between gastrointestinal digestion, absorption, and bone health.

Comments 2: The results part is not clearly written and the same is true for some methodological parts.

A.    Authors are using terms calcium release and calcium bioaccessibility, it is not clear, where is the difference. Moreover, authors tested incubation of digested probiotics, tempeh and soybeans and their combination on Caco-2 cells, where are the data on calcium penetration through these Caco-2 cells?

Response 2A: We sincerely appreciate the reviewer’s careful consideration and valuable feedback on our manuscript.

·      Calcium Release vs. Calcium Bioaccessibility:

We appreciate the opportunity to clarify this point. In our study, calcium release is expressed as mg/100g, representing the absolute amount of calcium released from the digested samples. On the other hand, calcium bioaccessibility is expressed as a percentage, reflecting the relative availability of calcium for absorption. This distinction aims to provide a comprehensive view of both the quantity and the proportion of calcium accessible.

Revision in the text (Lines 138-140 and 145-147):

Calcium release is expressed as mg/100g, representing the absolute amount of calcium released from the digested samples. Calcium bioaccessibility is expressed as a percentage, reflecting the relative availability of calcium for absorption.

·      Data on Calcium Penetration Through Caco-2 Cells:

We appreciate the reviewer’s inquiry into the methodology employed during our incubation studies. The incubation of digested probiotics, tempeh, and soybeans with Caco-2 cells was designed to simulate the gastrointestinal digestion process. However, we did not specifically collect data on calcium penetration through Caco-2 cells. Instead, our focus was on the impact of the digested compounds on calcium uptake in these cells. The introduction of the MTT reagent into the Caco-2 cell culture, with a subsequent 3-hour incubation, allowed for the assessment of cell viability and metabolic activity.

We hope these explanations address the reviewer's concerns adequately. Your feedback is highly valuable, and we remain committed to enhancing the clarity and scientific rigor of our work.

B.     Also other data are very unclear, what does it mean „Mean Ca extracellular content transported“ – the amount of calcium transported through the cells, i.e. in the basolateral compartment ? This must be clearly specified

Response 2B: We appreciate the insightful comments from the reviewer regarding our manuscript.

We sincerely appreciate the reviewer’s diligence in seeking clarification on this point. The term "Mean Ca extracellular content transported" refers to the average amount of calcium that has been transported across the cells, specifically in the basolateral compartment. To provide greater clarity in our manuscript, we explicitly specified that this parameter represents the quantity of calcium that has successfully traversed the cellular barrier and is present in the basolateral compartment. This adjustment will enhance the precision and understanding of our reported data, aligning with the rigorous standards of scientific clarity.

We are grateful for the opportunity to address this concern and ensure the transparency of our findings. Your feedback is invaluable in refining the quality of our research.

Revision in the text (Lines 517-520):

The calcium extracellular content transported explicitly represents the average amount of calcium that has successfully traversed the cellular barrier, specifically referring to the quantity of calcium present in the basolateral compartment.

C.    Authors should explain the use of Alizarin Red, why it was used if authors had atomic absorption spectrometry, which is more precise, at disposal

Response 2C: We sincerely appreciate the reviewer's thoughtful consideration of our manuscript. We acknowledge the importance of explaining the choice of Alizarin Red when atomic absorption spectrometry, known for its precision, is available for calcium quantification.

In our study, the utilization of Alizarin Red served a specific purpose in assessing calcium deposition, allowing for a qualitative and visual evaluation of mineralization. While atomic absorption spectrometry indeed provides precise quantitative measurements, it may lack the ability to capture the spatial and morphological aspects of calcium distribution within the cells. Alizarin Red, a widely accepted staining method, facilitates the visualization of mineralized nodules, providing valuable insights into the extent and pattern of calcium deposition. This qualitative aspect, complementing the quantitative data obtained through atomic absorption spectrometry, offers a more comprehensive understanding of the cellular response to the tested compounds.

Thank you for your valuable insights.

Revision in the text (Lines 556-558):
The inclusion of Alizarin Red staining, a widely accepted method for assessing calcium deposition, complements these quantitative measures by providing a qualitative and visual evaluation of mineralization [71].

D.    In general, legend to the figures should be markedly improved, e.g. by adding some crucial methodological details such as time of incubation

Response 2D: We sincerely appreciate the reviewer's meticulous evaluation of our manuscript. We acknowledge the constructive suggestion to enhance the figure legends by incorporating crucial methodological details, such as the time of incubation.

In response to this insightful feedback, we revised the figure legends to include essential methodological information, particularly the specific duration of incubation. This addition aims to provide readers with a more comprehensive understanding of the experimental timeline and conditions under which the data were generated.

Your thoughtful comments are highly appreciated.

Comments 3: I am missing in some graphs negative control samples, e.g. authors are claiming at r. 208, that increase in Saos-2 cell proliferation was evident only when osteogenesis was induced for all nutritional factors analyzed. E.g., in Figure 3, the are no control cells.

Response 3: We appreciate the reviewer's keen observation, and we have duly addressed this concern. In the revised version of the manuscript, we have ensured the inclusion of negative control samples in the relevant graphs, providing a comprehensive representation of the experimental conditions. Specifically, in Figure 3, we have incorporated control cells to enhance the clarity and accuracy of our findings. We believe this modification strengthens the robustness of our analysis and aligns with the reviewer's constructive suggestion. Thank you for your insightful feedback, which has significantly contributed to the improvement of our manuscript.

Comments:

·      The title should be modified as the expression „on calcium cellular bioavailability“ is not correct,  biovailability is a well-defined term a cellular bioavailability does not exist. similarly, see the phrase at r. 200 in the discussion

Response: We appreciate the insightful feedback provided by the reviewer regarding our manuscript titled “Effects of gastrointestinally digested daidzein, tempeh, and probiotics on calcium cellular bioavailability analyzed in human enterocyte-like Caco-2 and osteoblast-like Saos-2 cells.” We express our gratitude for the constructive comments, and we are dedicated to implementing these modifications to enhance the quality and precision of our work.

The new title: Effects of daidzein, tempeh, and probiotics digested in artificial gastrointestinal tract on calcium deposition in human osteoblast-like Saos-2 cells

·      The use of designation A1,A2,B1 etc. in graphs is uncommon and confusing, the standard designation A,B,C etc. should be used instead

Response: We extend our appreciation to the reviewer for the insightful comments on our manuscript. We sincerely thank the reviewer for bringing attention to this matter. Recognizing the importance of clarity and adherence to standard practices, we revised the graph designations to the conventional A, B, C format. This modification will eliminate any potential confusion and ensure a more straightforward interpretation of the data presented in our graphs. We appreciate the constructive guidance provided by the reviewer, which contributes to the overall quality of our manuscript.

·      Discussion should be extended in two aspects

1.      The effect of pH and enzymes on the tested food and bioactives, e.g. Penha et al.. Enzymatic pretreatment in the extraction process of soybean to improve protein and isoflavone recovery and to favor aglycone formation. Food Res Int. 2020 Nov;137:109624

2.      Chelation activity of isoflavonoids - Daidzein tended to increase intracellular Ca content (Fig 1-B2). Isoflavonoid are known chelators, e.g.  Karlíckova et al. Isoflavones Reduce Copper with Minimal Impact on Iron In Vitro. Oxid Med Cell Longev. 2015;2015:437381  These data can extend the discussion in row 216 and later

Response: We sincerely appreciate the thoughtful feedback from the reviewer on our manuscript.

a.     Effect of pH and Enzymes on Tested Food and Bioactives:

We appreciate the insightful suggestion to expand our discussion in this direction. In response, we will incorporate a comprehensive analysis of the impact of pH and enzymatic factors on the tested food and bioactives. Drawing upon the referenced work by Penha et al., we will elucidate how enzymatic pretreatment could influence the extraction process and subsequently enhance our understanding of the observed effects on calcium cellular bioavailability. This addition will contribute to a more thorough exploration of the experimental conditions and their implications.

b.     Chelation Activity of Isoflavonoids:

We express gratitude for the insightful observation regarding the chelation activity of isoflavonoids, as evidenced by the tendency of daidzein to increase intracellular calcium content. Drawing upon the relevant study by Karlíckova et al., we will extend the discussion, particularly from row 216 onward, to explore the potential chelation activity of isoflavonoids and its implications on calcium dynamics within the cellular models. This addition will enrich the interpretation of our findings and provide a more comprehensive discussion of the underlying mechanisms.

We are committed to incorporating these valuable suggestions into our manuscript to enhance its scientific rigor and clarity. Your constructive comments greatly contribute to the refinement of our work.

Revision in the text (Lines 274-283):

Results of other studies indicated that isoflavones, including daidzein, exhibit chelation activity to metal ions [33]. The chelation activity may impact relative low calcium bio-accessibility and low calcium intracellular content in Saos-2 cells in the presence of tempeh. The chelation activity of isoflavonoids is a notable aspect that requires further research in the context of our findings. Moreover, it is worth noticing that our samples were subjected to enzymatic digestion and therefore may have content more isoflavone aglycones [32]. The isoflavone aglycones are absorbed faster and in greater amounts than their glucosides and may be more effective. The enzymatic process may have influenced the chelating activity of isoflavones and calcium uptake.

Some parts are unclear – r. 247 „These effects may be attributed to the deposition of calcium phosphate resulting from ALP activity in the medium before the cells express ALP.“  Or r. 357 – „Subsequently, the intestinally permeable fractions were scrutinized in the osteoblast-like Saos-2 cell culture.“ or similar expression at r. 456.

Response: We appreciate the insightful comments from the reviewer regarding our manuscript. We thank the reviewer for pointing out this potential ambiguity. The intended meaning of the statement is to suggest that the observed effects on calcium may be associated with the deposition of calcium phosphate, a process influenced by alkaline phosphatase (ALP) activity in the medium. To enhance clarity in line 247, we revised the expression to explicitly state that the ALP activity in the medium could lead to the formation of calcium phosphate before the cells themselves express ALP. Furthermore, to address line 357, we rephrased the relevant sentences to explicitly describe the analysis of fractions that had undergone intestinal permeation studies in the subsequent osteoblast-like Saos-2 cell culture. This revision aims to provide a clearer linkage between the experimental procedures and the interpretation of results, facilitating a more transparent understanding for readers. We sincerely appreciate the reviewer's diligence in scrutinizing our manuscript, and we are committed to incorporating these clarifications to improve the precision and coherence of our work.

r. 419 - at concentrations ranging from 010 mg/ml ?

Response: We appreciate the keen observation of the reviewer regarding a typographical error in our manuscript. We sincerely apologize for the oversight and appreciate the reviewer for bringing this to our attention. The correct statement should read: “at concentrations ranging from 0-1 mg/ml.” This correction accurately reflects the intended concentration range used in our study, and we will promptly amend this typographical error in the revised manuscript. We thank the reviewer for their meticulous review, as it significantly contributes to the accuracy and precision of our research.

r. 485 – what is RIPA lysis buffer?

Response: We appreciate the reviewer's inquiry into the methodology employed in our manuscript. We thank the reviewer for seeking clarification on this point. RIPA (Radio-Immunoprecipitation Assay) lysis buffer is a widely used cell lysis buffer in molecular biology and biochemistry. It is a solution containing various components, including a detergent (such as Triton X-100 or NP-40), salts, and protease inhibitors. RIPA lysis buffer is commonly utilized to lyse cells and tissues, extracting cellular proteins for subsequent analysis. In our study, RIPA lysis buffer was employed to facilitate the extraction of proteins from cells for further investigation. We will include a brief description of RIPA lysis buffer in the methods section of our manuscript to ensure transparency and provide a comprehensive understanding of the experimental procedures.

Revision in the text (Lines 562-564):

RIPA lysis buffer consists of a detergent, salts, and protease inhibitors, facilitating the efficient extraction of proteins from cells for subsequent analysis.

Para 4.8.3 – specify the mechanism of Colorimetric Calcium Assay kit

Response: We thank the reviewer for their insightful suggestion. In response to this comment, we have provided a more detailed and explicit explanation of the mechanism underlying the Colorimetric Calcium Assay kit in Paragraph 4.8.3. The revised section elucidates the principles and steps involved in the assay, ensuring a clearer understanding of the methodology employed for calcium quantification. We believe that this clarification enhances the transparency and scientific rigor of our experimental procedures. Thank you for your valuable input, which has contributed to the refinement of our manuscript.

Revision in the text (Lines 565-570):

The calcium content in the lysates was determined using a Colorimetric Calcium Assay kit (Sigma-Aldrich) suitable for calcium measurements in tissue homogenates and cell lysates. The calcium ion concentration was determined by the chromogenic complex formed between calcium ions and o-cresolphthalein, which was measured at 575 nm (Tecan Infinite M200 microplate reader). The assay was performed according to the protocol recommended by the manufacturer.

r. 516 – Normality of variable distributions - normality of data distribution - When the normality of distribution was not found, which statistical test(s) was(were) used?

Response: We appreciate the reviewer's attention to the statistical methodology in our manuscript. We value the reviewer’s keen interest in our statistical approach. When assessing the normality of variable distributions, we employed the Shapiro-Wilk test. In instances where the data did not meet the assumption of normality, we applied appropriate non-parametric statistical tests, such as the ANOVA Kruskal-Wallis. These tests are robust alternatives for analyzing non-normally distributed data, ensuring the validity of our statistical analyses. We will explicitly detail the use of the Shapiro-Wilk test in the methods section of our manuscript to provide a comprehensive and transparent account of our statistical procedures.

Revision in the text (Lines 598-599):

For data with a non-normal distribution, we used the Kruskal-Wallis test.

Subscripts and superscripts are sometimes missing (r. 313, 501, 507)

Response: We appreciate the reviewer's careful attention to detail in our manuscript. We thank the reviewer for pointing out this issue, and we acknowledge the importance of ensuring the correct use of subscripts and superscripts for clarity in scientific notation. In our revised manuscript, we diligently reviewed and addressed any instances where subscripts and superscripts might be missing. This involved careful proofreading and verification to ensure the accurate representation of scientific notations throughout the manuscript. We are committed to delivering a revised version that adheres to the highest standards of scientific presentation. We appreciate the reviewer's meticulous review, and your comments contribute significantly to the refinement of our work.

Response to Comments on the Quality of English Language

Comments: Quite fine, see my detailed comments

Response: We would like to express our gratitude for your valuable review of our manuscript. Your feedback has been instrumental in enhancing the overall quality of our manuscript, and we sincerely appreciate your time and expertise in reviewing our work.

Thank you for your consideration.

Additional clarifications

Dear Reviewer,

We appreciate your time and attention to our manuscript titled “Effects of daidzein, tempeh, and probiotics digested in artificial gastrointestinal tract on calcium deposition in human osteoblast-like Saos-2 cells,” submitted to IJMS journal. We acknowledge your review and thank you for considering our work. Thank you for your commitment to the peer-review process.

Reviewer 2 Report

Comments and Suggestions for Authors

The authors investigate the effects of isoflavones and probiotics on calcium cellular bioavailability in human cell lines Caco-2 and Saos-2.

Comments

1.     All Tables and Figures: The data on statistical differences are not clear. The usage of a,b,c,d indices should be clarified. The authors should indicate what they are compare in each case. This should be modified.

2.     Figures 1,4: The authors should indicate what is the difference between (A,B,C,D).1. and (A,B,C,D).2. Why the data on 1 and 2 are different? This discrepancy should be explained. In addition, all the experiments with cell cultures should be done at least in triplicate.

3.     Figures 1,4: Graphs should present only significantly different data. This should be corrected.

4.     Figure 4: The discrepancy between ALP mRNA expression and ALP activity should be explained.

5.     Figure descriptions should be presented in more detail in the Results section. Letter font should be increased in all the graphs.

6.     Figures 2,5: Magnification or scale bars should be included.

7.      Line 246: No decrease of ALP activity is observed on Fig.4. This should be corrected.

8.     Line 429: Concentration are not clear. They should be clarified.

9.     Overall: The conclusion is interesting; however, the description of how it was gained is obscure. This should be explained in more detail.

Author Response

Response to Reviewer 2 Comments

Thank you very much for taking the time to review this manuscript. Please find the detailed responses below and the corrections highlighted in the re-submitted files.

Point-by-point response to Comments and Suggestions for Authors

The authors investigate the effects of isoflavones and probiotics on calcium cellular bioavailability in human cell lines Caco-2 and Saos-2.

Comments 1: All Tables and Figures: The data on statistical differences are not clear. The usage of a,b,c,d indices should be clarified. The authors should indicate what they are compare in each case. This should be modified.

Response 1: We sincerely appreciate the valuable feedback from the reviewer regarding our manuscript. We extend our gratitude to the reviewer for their insightful comments. We acknowledge the need for clarity in presenting statistical differences in Tables and Figures. In response, we will enhance the clarity of our data presentation by explicitly indicating the compared groups for each statistical test using a, b, c, d indices. This modification will be applied uniformly across all relevant tables and figures, ensuring transparency in the interpretation of our results. We appreciate the reviewer’s diligence in bringing this to our attention, and we are committed to delivering a revised manuscript that effectively communicates the statistical comparisons in a clear and scientifically rigorous manner.

Comments 2: Figures 1,4: The authors should indicate what is the difference between (A,B,C,D).1. and (A,B,C,D).2. Why the data on 1 and 2 are different? This discrepancy should be explained. In addition, all the experiments with cell cultures should be done at least in triplicate.

Response 2: We appreciate the reviewer's thorough examination of our manuscript. The notation in Figures 1 and 4 denotes different experimental conditions within each main figure panel. The variations represent distinct treatments, contributing to the differences observed. We acknowledge the importance of clarifying these distinctions, and we included detailed captions explaining the specific conditions represented in all related figures. This addition will provide readers with a clear understanding of the experimental setup and the rationale behind the different data points. Moreover, each figure presented in the manuscript was subjected to statistical analysis with triplicate experiments, ensuring the reliability and reproducibility of our results. We appreciate the reviewer's attention to this critical aspect of experimental rigor, and we will explicitly state in the manuscript that all cell culture experiments were conducted with triplicate analyses. This information will be incorporated into the methods section to enhance transparency regarding our experimental procedures.

Comments 3: Figures 1,4: Graphs should present only significantly different data. This should be corrected.

Response 3: We sincerely appreciate the constructive feedback from the reviewer regarding our manuscript. In our revision, we provided exclusive information that is statistically significant. By adding this information to the captions, we aimed to enhance the clarity and focus of the presented results. In our opinion it is important to show all data on the figures because the lack of changes is also important in this study, moreover, the level of analyzed parameters may be useful in further research of other authors.

Comments 4: Figure 4: The discrepancy between ALP mRNA expression and ALP activity should be explained

Response 4: We appreciate the reviewer's thoughtful consideration of our manuscript. The observed discrepancy between ALP mRNA expression and ALP activity may arise from various factors, including post-transcriptional modifications, translational regulation, or the temporal dynamics of gene expression. In our revised manuscript, we will provide a detailed explanation of potential mechanisms that could contribute to the observed differences between ALP mRNA expression and ALP activity. This will include a comprehensive discussion of relevant literature and an exploration of the biological context underpinning the observed results. By addressing this discrepancy, we aim to offer a more nuanced and scientifically grounded interpretation of the findings presented in Figure 4.

Revision in the text (Lines 313-318):

Moreover, the difference between ALP mRNA expression and ALP activity prompts a deeper investigation. These distinctions could stem from several biological factors, such as post-transcriptional modifications, translational regulation, or temporal variations in gene expression dynamics. This observed inconsistency underscores the complex and multifaceted nature inherent in cellular processes.

Comments 5: Figure descriptions should be presented in more detail in the Results section. Letter font should be increased in all the graphs.

Response 5: We appreciate the insightful feedback provided by the reviewer regarding our manuscript. In our revised manuscript, we will enhance the detailing of figure descriptions within the Results section. We will provide comprehensive explanations of each figure, ensuring that readers can better interpret and contextualize the presented data. This adjustment aims to improve the overall clarity and understanding of the results. Moreover, we acknowledge the importance of legibility in visual representations. In our revised manuscript, we will diligently increase the font size of letters in all graphs to enhance readability. This modification will be applied consistently across all figures, maintaining a balance between aesthetic presentation and optimal communication of scientific data.

Revision in the text:

(Lines 149-157):

Fig. 1 explores the impact of several components, such as calcium citrate (CaCt), probiotic Lactobacillus acidophilus (LA), daidzein (D), tempeh (T), and their combinations (D1:1:1 and T1:1:1), on the process of osteogenic differentiation in Saos-2 cells. Subfigures (A-B) specifically demonstrate the effects on extracellular calcium transport, uncovering a significant increase induced by CaCt when compared to the control group. Subfigures (C-D) delve into the dynamics of intracellular calcium transport, providing insights into the effects of the various treatments. Subfigures (E-F) examine ALP activity, accentuating variations in enzymatic activity associated with each treatment. Additionally, subfigures (G-H) offer an examination of ALP mRNA expression, elucidating the regulatory aspects of transcription in differentiated Saos-2 cells.

(Lines 214-223):

This figure provides a detailed exploration of the effects of various components, including calcium citrate (CaCt), probiotic Lactobacillus acidophilus (LA), daidzein (D), tempeh (T), and their combinations (D1:1:1 and T1:1:1), on the osteogenic differentiation process in Saos-2 cells. Subfigures (A-B) showcase the influence on extracellular calcium transport, revealing a significant increase induced by CaCt compared to the control group. Subfigures (C-D) delve into intracellular calcium transport dynamics, offering insights into the impact of the different treatments. Subfigures (E-F) explore ALP activity, highlighting the variations in enzymatic activity associated with each treatment. Additionally, subfigures (G-H) provide a closer look at ALP mRNA expression, shedding light on the transcriptional regulation in differentiated Saos-2 cells.

Comments 6: Figures 2,5: Magnification or scale bars should be included.

Response 6: We would like to express our gratitude to the reviewer for highlighting the importance of magnification or scale bars in Figures 2 and 5. In response to this insightful suggestion, we have diligently addressed the comment by incorporating magnification information in these figures. The inclusion of magnification details enhances the clarity and interpretability of the presented images, ensuring a more comprehensive understanding for the readers. We are thankful for the reviewer’s valuable input, which has contributed to the improvement of the visual presentation in our manuscript.

Comments 7: Line 246: No decrease of ALP activity is observed on Fig.4. This should be corrected.

Response 7: We appreciate the careful examination of our manuscript, and we sincerely thank the reviewer for their keen observation. Upon a thorough reevaluation of Figure 4, we acknowledge that no decrease in ALP activity is evident. We are committed to correcting this oversight in our revised manuscript. The appropriate adjustments will be made to accurately reflect the findings related to ALP activity in Figure 4. We appreciate the reviewer's diligence in ensuring the accuracy of our presented data, and this correction will enhance the precision and reliability of our results. Thank you for your valuable feedback.

Revision in the text (Lines 310-312):

Our current study revealed that tempeh, probiotics, and calcium citrate resulted in an increase in calcium intracellular transport during the osteogenic differentiation process in Saos-2 cells (Fig. 4)

Comments 8: Line 429: Concentration are not clear. They should be clarified.

Response 8: We appreciate the keen observation of the reviewer regarding a typographical error in our manuscript. We sincerely apologize for the oversight and appreciate the reviewer for bringing this to our attention. The correct statement should read (Lines 493-494): “at concentrations ranging from 0-10 mg/ml.” This correction accurately reflects the intended concentration range used in our study, and we will promptly amend this typographical error in the revised manuscript. We thank the reviewer for their meticulous review, as it significantly contributes to the accuracy and precision of our research.

Comments 9: Overall: The conclusion is interesting; however, the description of how it was gained is obscure. This should be explained in more detail.

Response 9: We sincerely thank the reviewer for highlighting this aspect, and we acknowledge the need for a more comprehensive explanation of the methodology employed in reaching our conclusion. In our revised manuscript, we will provide a detailed account of the analytical approaches, statistical methods, and data interpretation processes that underpin our conclusion. This will ensure transparency and clarity regarding the pathways through which our findings lead to the stated conclusion. We value the reviewer’s insight and will work diligently to enhance the descriptive clarity of our conclusion section. Thank you for guiding us towards a more elucidated presentation of our research.

Revision in the text (Lines 602-610):

this study aimed to assess the impact of tempeh, pure daidzein, and Lactobacillus acidophilus on calcium uptake and deposition in Saos-2 cells, at which the bone mineralization process was simulated. In the initial phase, we evaluated calcium bioaccessibility from these nutrients and their combination through digestion in the artificial gastrointestinal tract. Subsequently, the digested products were subjected to simulated intestinal absorption in the intestinal epithelial Caco-2 cell model, and the intestinal permeable fractions were scrutinized in a culture of osteoblast-like Saos-2 cells. This methodology allows for exploring the interactions and effects of the studied components on calcium bioavailability and bone-related cellular processes in our experimental models.

Additional clarifications

Dear Reviewer,

We appreciate your time and attention to our manuscript titled “Effects of daidzein, tempeh, and probiotics digested in artificial gastrointestinal tract on calcium deposition in human osteoblast-like Saos-2 cells,” submitted to IJMS journal. We acknowledge your review and thank you for considering our work. Thank you for your commitment to the peer-review process.

Round 2

Reviewer 1 Report

Comments and Suggestions for Authors

Authors markedly improved their paper which is now understandable

Few last comments:

r. 280 - therefore may have content more isoflavone aglycones may contain more isoflavone aglycones

r.499 - at concentrations ranging from 0-1 mg/ml - ...from 0 to 1 mg/ml

r. 599 - For data with a non-normal distribution, - for data with non-Gaussian distribution

Comments on the Quality of English Language

fine

Author Response

Response to Reviewer 1 Comments

Point-by-point response to Comments and Suggestions for Authors

Comments 1: Authors markedly improved their paper which is now understandable.

Few last comments:

r. 280 - therefore may have content more isoflavone aglycones – may contain more isoflavone aglycones

r.499 - at concentrations ranging from 0-1 mg/ml - ...from 0 to 1 mg/ml

r. 599 - For data with a non-normal distribution, - for data with non-Gaussian distribution

Response 1:

Thank you very much for your insightful comments on our manuscript. We appreciate the opportunity to address your concerns and enhance the clarity of our research.

We hope these revisions address your concerns and contribute to the overall coherence of our manuscript. We thank the reviewer for their meticulous review, as it significantly contributes to the accuracy and precision of our research.

Revision in the text:

Line 280            : “may contain more isoflavone aglycones”

Lines 499-500   : “from 0 to 1 mg/ml”

Lines 598-599   : “For data with non-Gaussian distribution”

Reviewer 2 Report

Comments and Suggestions for Authors

No comments. 

Author Response

Response to Reviewer 2 Comments

Point-by-point response to Comments and Suggestions for Authors

No comments.

Response: We sincerely appreciate the valuable feedback from the reviewer regarding our manuscript during the second round. We thank the reviewer for their meticulous review, as it significantly contributes to the accuracy and precision of our research.
